# A COVID-19 Outbreak in a Large Meat-Processing Plant in England: Transmission Risk Factors and Controls

**DOI:** 10.3390/ijerph20196806

**Published:** 2023-09-22

**Authors:** Yiqun Chen, Helen Beattie, Andrew Simpson, Gillian Nicholls, Vince Sandys, Chris Keen, Andrew D. Curran

**Affiliations:** Science Division, Health and Safety Executive, Buxton SK17 9JN, UK; helen.beattie@hse.gov.uk (H.B.); andrew.simpson@hse.gov.uk (A.S.); gillian.nicholls@hse.gov.uk (G.N.); vince.sandys@hse.gov.uk (V.S.); chris.keen@hse.gov.uk (C.K.); andrew.curran@hse.gov.uk (A.D.C.)

**Keywords:** COVID-19, SARS-CoV-2, food manufacturing, workplace, outbreak, risk factor, control measure, environment, night-shift work, ventilation

## Abstract

The meat-processing industry had frequent COVID-19 outbreaks reported worldwide. In May 2021, a large meat-processing plant in the UK had an outbreak affecting 4.1% (63/1541) of workers. A rapid on-site investigation was conducted to understand the virus transmission risk factors and control measures. This included observational assessments of work activities, control measures, real-time environmental measurements and surface microbial sampling. The production night-shift attack rate (11.6%, 44/380) was nearly five times higher than the production day-shift (2.4%, 9/380). Shared work transport was provided to 150 staff per dayshift and 104 per nightshift. Production areas were noisy (≥80 dB(A)) and physical distancing was difficult to maintain. Face visors were mandatory, additional face coverings were required for some activities but not always worn. The refrigeration system continuously recirculated chilled air. In some areas, the mean temperature was as low as 4.5 °C and mean relative humidity (RH) was as high as 96%. The adequacy of ventilation in the production areas could not be assessed reliably using CO_2_, due to the use of CO_2_ in the packaging process. While there were challenges in the production areas, the observed COVID-19 control measures were generally implemented well in the non-production areas. Sixty surface samples from all areas were tested for SARS-CoV-2 RNA and 11.7% were positive. Multi-layered measures, informed by a workplace specific risk assessment, are required to prevent and control workplace outbreaks of COVID-19 or other similar respiratory infectious diseases.

## 1. Introduction

Severe acute respiratory syndrome coronavirus 2 (SARS-CoV-2) is a highly transmissible virus that has caused a pandemic of coronavirus disease (COVID-19). SARS-CoV-2 virus can spread directly or indirectly from person to person primarily through respiratory droplets and aerosols, and can be spread by asymptomatic, pre-symptomatic and symptomatic individuals [1]. Outbreaks of COVID-19 in meat-processing plants have frequently been reported from many parts of the world [2,3,4,5,6,7,8,9], causing temporary plant closures, labour shortages and supply chain disruptions [10]. In the UK in 2021, the meat-processing sector employed around 97,000 people [11], and produced over 4 million tonnes of meat valued at £9 billion [12]. This sector continued to operate throughout the pandemic, including during national lockdowns, with the workforce classified as ‘essential workers’. It is important to investigate the risk factors of COVID-19 outbreaks in this sector to inform effective control measures and to support future pandemic preparedness.

Combinations of factors are likely to contribute to the frequency of workplace outbreaks in the meat-processing sector. Some of these are unique to the meat-processing sector, including a high density of workers and environmental conditions that may favour virus transmission, such as recirculating chilled air, lack of outside fresh air and low temperatures. Some studies also investigated the potential contribution of social inequalities in this sector, including foreign-born workers, employment contract type and shared accommodation and work commute [13]. Although published studies suggested inhalation transmission was particularly important [14] and deboning and meat cutting production areas were particularly at high risk [4,5], additional measurements were needed to evaluate the roles of environmental conditions and ventilation on the transmission risk in these workplaces [15].

This study describes an investigation, as part of the COVID-OUT (COVID-19 Outbreak Investigation to Understand Transmission) study [16], of an outbreak in a large meat-processing plant. It details the investigation methods and findings of the comprehensive environmental assessment, surface microbial sampling and analyses of local and public health data. The aims of the study were: (1) to understand the viral transmission risk factors, potential transmission routes and their contribution in this COVID-19 outbreak; and (2) to identify and evaluate the range of measures, including workplace testing and contact tracing, physical distancing, face coverings, screens, ventilation and cleaning, implemented in this workplace to prevent and control the outbreak.

## 2. Materials and Methods

This outbreak investigation in a large meat-processing plant was part of the COVID-OUT study that involved on-the-ground investigations of twelve COVID-19 outbreaks in a range of workplace settings including manufacturing [17,18], warehousing and distribution and public-facing offices [19]. These investigations used the previously published COVID-OUT study protocol [20] to systematically evaluate workplace COVID-19 outbreaks as soon as reasonably practicable when an outbreak was identified.

The COVID-OUT study protocol was designed using a set of relevant World Health Organization (WHO) COVID-19 early investigation protocols [21,22,23,24] which were combined and adapted for the investigations of workplace outbreaks in the UK. It has a general data framework and data collection tools developed based on learnings from previous responses to global emerging pathogens [25,26]; it considers individual, environmental and population level data needed to achieve the study objectives.

The protocol was applied consistently to the COVID-OUT study across multiple outbreak sites, including this investigation at a meat-processing plant. The methods are described in more detail as follows.

### 2.1. Outbreak Identification and Recruitment

A system was implemented where suitable outbreak sites were identified and notified to the study team by Public Health England (PHE, which became the UK Health Security Agency (UKHSA) on 1 October 2021), the Health and Safety Executive (HSE), local authorities or individual organisations. Upon notification, the research team would rapidly approach and recruit companies that were experiencing outbreaks. With the employers’ informed consent, the research team would visit the outbreak site as early as feasible to collect relevant data. Participation in the study was voluntary. The COVID-OUT study was approved by the NHS North East Research Ethics Committee (Reference 20/NE/0282).

On 9 June 2021, a workplace outbreak of COVID-19 was reported to the COVID-OUT study team. The outbreak was located in a large secondary meat-processing plant in England that processes (including cutting/deboning, packaging) and dispatches over 1.5 million meat products per week for food retail. The company provided informed consent for the environmental assessment part of the study. This included allowing the research team access to the workplace to assess the physical environment, work practices and COVID-19 control measures; to conduct surface microbial sampling; and to access and analyse anonymised data relevant to this research study. The data included information on the workforce, work shifts, workplace testing, sequence of confirmed COVID-19 cases identified, and the company COVID-19 risk assessment.

### 2.2. Case Definition and Identification

A confirmed case was defined as an individual who worked in the plant and had a positive polymerase chain reaction (PCR) test or rapid antigen test by lateral flow device (LFT) for SARS-CoV-2 during the outbreak period. Anonymised information about positive cases, including dates of the positive tests, obtained from PHE and the company was combined. The PHE case list contained information on the number of employees who tested positive from 29 May 2021 up to 15 June 2021, including symptom onset dates, follow-up test dates, work shifts, whether working from home, linked positive tests in family members and S-gene positive status of the PCR test samples. A detectable S-gene was a proxy for identifying Delta variant of SARS-CoV-2 infection after April 2021 in England. The company’s case list contained information on the number of employees who tested positive from 3 January 2021 to 27 June 2021, including information on test dates, work shifts, job title and work areas. When the two case lists were compared, there were 11 cases (seven in nightshifts, two in dayshifts and two without information on shift type) in the PHE case list that did not appear in the company case list. These cases were added to the company case list to provide the final combined case list. The outbreak dynamics were illustrated using an epidemic curve, produced using Stata statistical software 17 [27]. Attack rates were calculated by dividing the number of cases by the total number of workers [28], using workforce data provided by the company. The calculation was undertaken for the whole plant, and by work shift.

### 2.3. Environmental Assessment, Including Real-Time Environmental Measurements

The on-site environmental assessment and surface sampling were carried out on 24 and 25 June 2021 when the outbreak was declining but there were still occasional positive cases identified in the workforce. The environmental assessment was conducted following previously developed data collection frameworks and tools [29], which included collecting information on the building layout, ventilation, temperature, humidity, air movements and noise levels, the workforce (e.g., the number of staff and shift patterns) and the observed work activities (e.g., worker adherence to infection control measures and worker interactions).

Spot measurements for carbon dioxide (CO_2_) (as a proxy for ventilation), relative humidity and temperature were taken on the two-day site visit, as well as longitudinally over two weeks, from 25 June to 11 July 2021. The real-time monitors (CO_2_: Honeywell BW Solos; temperature and humidity: EL-SIEs) were left in various locations around the plant where workers may congregate. They were positioned at around head height at locations that would avoid direct worker interaction and away from ventilation grilles. Data were logged every minute and a moving 30-min average applied. Honeywell BW Solo CO_2_ monitors have a resolution of 100 ppm, an estimated limit of detection of around 300 ppm, and an upper limit of 50,000 ppm. CO_2_ concentration levels can be used as a surrogate to indicate the adequacy of ventilation provision in occupied indoor spaces providing there are no other CO_2_ sources in the space. In multi-occupant spaces that are used regularly, a CO_2_ concentration routinely greater than 1500 ppm or a ventilation rate below 5 l/s/person is considered indicative of inadequate ventilation [15,30]. Real-time measurement results for CO_2_, relative humidity and temperature were analysed using Microsoft Excel version 2302.

### 2.4. Surface Microbial Sampling

For surface sampling for SARS-CoV-2 RNA, samples were collected from high touch points and other areas of interest (areas where workers congregated and workplaces throughout the site) using previously described methods [17,31]. The sampling used either Envirostik with Neutralising Buffer (Technical Service Consultants, TS/15-SH or TS/15-B) or UTM^®^ swabs (Copan, 366C). Collected samples were sent to laboratories at PHE Porton for analysis. Quantitative reverse transcription-polymerase chain reaction (qRT-PCR) for both ORF1ab and nucleocapsid (N) gene targets was performed in duplicate using the VIASURE SARS-CoV-2 Real Time PCR Detection Kit (CerTest, VS-NC02) in accordance with the manufacturer’s instructions. The reportable detection limit was a crossing threshold (Ct) value of 38.0 as recommended by the manufacturer’s instructions. Positive samples reported in analysis included confirmed positive samples and suspected positive samples. Confirmed positive samples were those with both replicates testing positive for at least one target, and suspected positive samples were those with a single replicate testing positive for at least one target. Positive samples with cycle threshold (Ct) values of 35.0 or lower were sent for further whole genome sequencing (WGS).

## 3. Results

The first confirmed case linked to this outbreak was identified on 29 May 2021. Case numbers then increased, leading to an outbreak being declared on 8 June 2021 by PHE. This time-period corresponded with the emergence of the Delta variant (B.1.617.2) of SARS-CoV-2 in the UK and this outbreak occurred as the local community infection rate started to increase (Figure 1). The rolling seven-day case rate in the surrounding lower tier local authority area (LTLA) was approximately 20 per 100,000 populations at the start of this outbreak. It increased ten-fold to around 200 per 100,000 by the end of June [32]. A preliminary investigation by PHE identified Delta variant in 16 of the first 25 cases of this outbreak. Information on variant types of the remaining cases was not available (Figure 2).

### 3.1. The Outbreak and the Attack Rates

The meat-processing plant employed 1541 workers who worked in dayshifts, nightshifts, or a combination of these across multiple work areas, such as processing, hygiene, quality assurance, engineering, despatch and transportation. Some worked part-time. Most of the workers were permanent employees with a small number (134, 8.7%) of agency staff and contractors. About a quarter (24.2%) of the workers were aged 50 or over. There were more men (56.8%) than women (43.1%). There were two work shifts, a dayshift (07.00–16.00, Monday–Friday) and a nightshift (21:30–06:30, Sunday–Thursday). Saturday and Sunday daytime were non-operational periods. Between the dayshift and the nightshift, there was a production hygiene shift (15:30–22:00) to clean the whole production area.

Approximately 380 staff members were working on production, either in a day or a nightshift. However, the nightshift had generally a more consistent and constant workforce. A company coach service was available, transporting on average 150 workers per dayshift and 104 workers per nightshift.

During the outbreak period, a total of 63 confirmed cases were identified, with an estimated overall attack rate of 4.1% (63/1541). A timeline of confirmed cases is shown in Figure 2. The estimated attack rate for the production night-shift workers (11.6%, 44/380) was almost five-fold higher than that of the production day-shift workers (2.4%, 9/380). For the remaining 10 confirmed cases, six worked both day and nightshifts and four did not provide information on their shift type. Four night-shift cases and a case without information on shift type self-reported positive test results among their family members within two days of their own positive result.

The first confirmed case was identified in a night-shift worker on 29 May, with a further seven cases reported in the following eight days. The first case reported in a day-shift worker was on 7 June. This case was a partner of a night-shift case who tested positive three days earlier (Figure 2, N(1)* and D(1)*).

### 3.2. Observational Environmental Assessment Results

The plant had approximately 15,000 m^2^ internal floor space, largely consisting of two chilled production areas (A and B; Figure 3) that contained large mechanical equipment for semi-automated product lines with overhead conveyors. Offices were situated on the first floor of the bottom end of production area A. Locker rooms were situated on the bottom end of production area B with canteen areas on the floor above. Additional locker rooms and smoking areas were provided outside to allow sufficient spacing.

Before the outbreak, the company had implemented temperature screening for all staff, visitors, contractors and delivery drivers entering the site and recommended twice weekly LFTs for all staff. An internal track and trace system was in place and led by senior managers and directors. The company offered full pay for COVID-19-related self-isolation. Staff communication boards were set up in various languages.

The company provided additional changing facilities, lockers, car parking space and amenities including canteen space and toilets to assist social distancing. Staff members in the office were also encouraged to work from home where possible. A one-way system had been trialled in some crowded areas but it was reported that this was not always followed. Break times, or shift start and end times, were staggered and were patrolled by management to ensure social distancing. There were signs displayed to reinforce the social distancing and no overcrowding was observed in the communal areas during the site visit. Face coverings were mandatory for all staff in communal areas (except when eating and drinking) and when travelling on the company coaches. Face coverings here included reusable fabric face masks provided by the company to workers. Workers were also allowed to use their own face coverings of any type. All types of face coverings were taken home to wash, and were not included in the general workwear laundry facility on-site, which provided washing for general production area coats and trousers. Supervision was provided for enforcing face coverings in these areas, and the observed adherence was good.

In the production areas, physical distancing could not always be maintained. There were crowded areas where production lines were close together, and people walked past in close proximity (<2 m). It was not always possible to place workers back-to-back or side-by-side rather than face-to-face when they worked closer than 2 m. Screens were erected between those working closely together but this was not always possible because of the production process or the production line configuration. In these situations, workers were required to wear both a face covering and a face visor, but it was observed that face coverings were not always worn in such situations.

The production areas were mandatory hearing protection zones. Noise levels in the meat material intake, packing and the main production areas were recorded at 80–83 dB(A) (Appendix A). A combination of factors, such as high noise level, using hearing protection and face coverings, together with some employees who did not have English as their first language which might have affected verbal communication, made it difficult to maintain >2 m physical distancing. Production staff members were observed having regular interactions with each other and with staff who were based in the office area (e.g., managers and engineers). The 2 m physical distancing was not maintained at all times during these interactions.

As part of the meat-processing activities, personal protective equipment in production areas included gloves, aprons, gauntlets and beard snoods where needed. For COVID-19 controls, personal visors were mandatory in the production areas and face coverings were required only if physical distancing was not reasonably practical. However, engineers would wear a face covering in production areas as visors were impractical for their work roles.

The factory was sanitised and deep cleaned at regular intervals during each 24-h period. Frequent hand washing was part of standard food hygiene requirements and had been reinforced through various communication platforms. Numerous hand sanitiser stations had been installed across the site. Signs to support the hand hygiene process were displayed above the sinks.

A number of non-production areas were fogged daily, Sunday to Thursday. Fogging was undertaken using Safe Zone Plus A006 EV viricidal disinfectant, containing 2-aminoethanol (0.1–1%), didecyldimethylammonium chloride (DDAC) (0.1–1%) and potassium carbonate (0.1–1%). The manufacturer’s information for this product stated that it was “effective against all enveloped viruses, including Coronavirus”, but it did not specifically state SARS-CoV-2. A spreadsheet containing details of fogging frequency and times showed that fogging was undertaken extensively, including once daily in the main office block; twice daily in the reception area, engineering workshop and logistics office; and three times daily in the canteen, locker rooms and toilets.

Soon after the declaration of the outbreak, on-site surge PCR testing was implemented for one week, between 13 and 20 June, testing 500 out of 860 staff on-site. The voluntary LFTs were stopped during this period and were reinstated on 20 June. Vaccination clinics were set up on-site and approximately 100 workers received vaccinations, including second doses. Additional coaches (increased from four to eight) were deployed to allow physical distancing in shared work transportation.

### 3.3. Measurement Results for Ventilation (CO_2_), Temperature and Humidity

In the non-production areas, there were two mechanical ventilation systems and some natural ventilation. The air handling units in these areas (Figure 3, AHU/NP1 and AHU/NP2) had G4 rated filters and provided 100% fresh air. Air conditioning units, with a combination of fresh and recirculated air, were also present in these areas. There were numerous vents in the ceiling in the offices and canteen areas. The main locker room had four ceiling vents at intervals in a row down the centre of the room. The temporary locker building (two-storey) was naturally ventilated by open doors and windows with no mechanical ventilation present.

In the production areas, there was one mechanical ventilation system (Figure 3, AHU/PA) and a refrigeration system. The air handling unit in production area A (AHU/PA) was an automatic supply ventilation system with balanced grilles, panel and bag filters, which were G4 and F9 rated, respectively, and provided 100% fresh air. It drew external air and chilled it to a set room temperature. It was designed to maintain a positive air pressure in the building and to comply with recommended air make-up rates of the system. A refrigeration system was operating in production area A, which was designed to keep the building and products cool. In addition, 21 air conditioning units kept chilled air continuously recirculated. There was no mechanical extraction of air from this area. The system was not designed to remove anything except heat and operation-generated moisture. The chilled air migrated from production area A into production area B via two link tunnels. The tray wash area in production area B had an air extraction system. The ’Individually Quick Frozen’ tunnel room required a chilled air make-up to compensate for the air extraction system to prevent build-up of nitrogen.

The ventilation and refrigeration systems operated continuously, and were controlled by building management systems which were maintained by two separate external contractors. One contractor was responsible for the systems in the production area and the other for the non-production area. Company staff could not alter the controls. The company COVID-19 risk assessment only covered the ventilation systems in the non-production area, and stated that the ventilation was adequate, used correctly and maintained/checked. However, the risk assessment did not consider the ventilation for the production areas and the information on air flow rates in the production areas was unavailable.

Spot measurements of CO_2_ concentrations were found to be elevated (≥1500 ppm) in the production areas (Appendix A) but it was identified after the measurements that CO_2_ was used to inhibit bacterial growth as part of the meat packaging process in these areas. In the main production area A, there were two main packing machine types that used CO_2_, including ten tray sealers (six of Mondini and four of Proseal) and nine vertical baggers (Ilapack VFFS). It was reported that the packing machine CO_2_ works on demand during production and is not switched off at the main inlet at the end of the shift. The CO_2_ data from the main production area and areas connected to it cannot therefore be used to assess the adequacy of ventilation in these areas. None of the areas away from the main production area had levels greater than 1500 ppm.

Real-time continuous CO_2_ measurements were also carried out over two weeks to assess ventilation (Appendix A). CO_2_ measurements were taken in the cut-preparation production area, which was the first part of the meat processing. Meat materials were received and cut up on a bench and then put on a conveyor into the main production area A. Although the CO_2_ level often exceeded 3000 ppm in this area, because it was connected to the main production area, separated only by flexible plastic drapes, the CO_2_ data from this area could not be used to assess the adequacy of ventilation, as previously described (Appendix A). The same might also explain the high CO_2_ levels (mean: 1126 ppm; range: 400–2500 ppm) in the washing area above sinks at the office-production entrance (Appendix A).

CO_2_ readings were used to assess the adequacy of ventilation in the non-production areas, as these should not be affected by the external sources of CO_2_ from the packaging process in the production areas. The real-time measurements of CO_2_ levels in the canteen (mean: 388 ppm; range: 300–800 ppm) and the main locker room (mean: 516 ppm; range: 300–1400 ppm) were within normal ranges. Figure 4 shows the continuous readings of CO_2_ level in the main locker room. The mid-week CO_2_ levels remained below 1500 ppm. There was generally a peak of short duration at the beginning and end of each shift. There were also broader peaks within the shift period.

Real-time measurements of temperature and humidity were also carried out over two weeks in both production and non-production areas (Appendix A). In general, the observed readings were in line with expected levels for the specific work areas. The cut-preparation area had a very low mean temperature of 4.5 °C (range 2.3–11.9 °C) and a very high mean relative humidity (RH) of 95.9% (range 67.5–100%). The temperature changes in this area aligned to the production days and times, with temperature around 3 °C to 4 °C during working shifts, and peaked between 9 °C and 12 °C (between 15:15 and 20:15) when the hygiene team were operating between shifts (Appendix A).

### 3.4. Surface Microbial Sample Results

A total of 60 surface samples were collected across the site on 24 and 25 June to assess the level of viral contamination. Only one sample was confirmed positive for SARS-CoV-2 RNA (1.7%, 1/60) and six samples (10.0%, 6/60) identified as suspected positive (Table 1). The engineering workshop had the highest concentration of samples with evidence of SARS-CoV-2 RNA, with one positive and two suspected positive in this location. This area was occupied by numerous engineers including external contractors. While most engineers had dedicated tools, there were high-touch multi-user items in this location. Only a single sample collected on site was confirmed positive, with a Ct value of 34.0. This sample was subjected to WGS analysis, but the sample did not pass quality control parameters for further analysis.

## 4. Discussion

This investigation of a COVID-19 outbreak at a large meat-processing plant identified many potential transmission risk factors, including high occupancy level, low temperature, high humidity, high noise levels, night-shift work and shared work transportation which may have contributed to cases detected in this workplace (Figure 5). Most of these were consistent with findings of previously published studies [5,14]. In addition, our investigation also identified challenges in the implementation of control measures, such as ventilation and face coverings, particularly in the production areas of this outbreak site. On the other hand, some other transmission risk factors indicated in previous studies were not present in this outbreak site, such as employment or socio-demographic factors, precarious work, on-site shared accommodation, not self-reporting COVID-19 symptoms or limited sick pay when staff self-isolate [4].

### 4.1. Context

Before the start of the outbreak at this site, England moved to further easing of COVID restrictions, as infection rates continued to decrease in the community, but COVID-secure rules for workplace and businesses remained [33]. The outbreak at this site started as the infection rate started to increase in the local community when the Delta variant became prevalent. Although the initial infection case(s) were introduced to the workplace from the community, it is possible that work and work-related risk factors might have contributed to the further spread of the virus. Findings of our environmental assessment were consistent with suggestions from an earlier study that a 2-m physical distance might not be sufficient to prevent virus transmission in the production areas of meat-processing plants, where there is a constant recirculation of chilled air and low fresh air exchange [14]. The distances at risk of transmission may vary greatly depending on the layout of the production areas, work activities, the occupancy level and fresh air supplies.

In general, there are three major routes where SARS-CoV-2 transfer from an infected person to a susceptible individual through respiratory droplets and aerosols: (1) the inhalation (airborne) route, which can occur at any distance but is more likely when people are in close proximity; (2) the spray (droplet) route, which usually occurs in close proximity; and (3) the touch (fomite) route [1]. All three routes may have played a part in the virus transmission in this outbreak. Our study could not assess the relative importance of the different transmission routes at this site, but it has identified the challenges in controlling the potential inhalation transmission in this outbreak site, particularly in the production areas.

### 4.2. Noise

Production lines are designed for maximum efficiency and productivity, with workers working closely together where physical distance is difficult to maintain at all times. Vocal effort for conversation at a distance of 1 m has to increase when ambient noise levels are above 65 dB(A) [34]. Previous studies have suggested that high levels of vocal effort in noisy environments could contribute to inhalation transmission of SARS-CoV-2 in workplace settings where physical distancing was limited [7,35]. In the production areas of the outbreak site, the ambient noise levels were all above 80 dB(A). The noisy production areas would have caused people to talk loudly, which might increase the aerosolization of the respiratory virus by infected workers.

### 4.3. Ventilation

The ventilation systems for the production areas and non-production areas were maintained by two different external contractors. The company COVID-19 risk assessment did not provide information on the ventilation assessment in the production areas. High levels of CO_2_ were found in the production areas, with many exceeding 1500 ppm. However, we could not determine if the high CO_2_ readings were from the presence of CO_2_ from the packaging process or from the lack of ventilation. The amount of CO_2_ being released from the packing process to the workplace air was not assessed by the study. The high level of CO_2_ in the production areas may also imply a decreased pH level, which could possibly increase SARS-CoV-2 stability in the air and therefore the transmissibility of the virus [36]. Such findings highlight the importance of detailed assessments to identify poorly ventilated work areas [37].

### 4.4. Temperature and Humidity

SARS-CoV-2 is sensitive to heat but is relatively stable at 4 °C [38]. Analysis of data from a large number of viral survival experiments in the air, including data for SARS-CoV-2, influenza viruses, MERS-CoV viruses and virus substitutes, has demonstrated that the lower the temperature, the greater the virus’s survival. Data at room temperature indicate that viruses also survive better in the air with lower (dry) or higher (humid) RH, but they survive less when the RH is between 40% and 60% [39]. Models also suggest that transmission risk from surfaces depends on environmental conditions, with high humidity facilitating greater transfer of virus from surfaces to hands [40]. Our environmental measurements found a very low temperature (mean 4.5 °C but could be as low as 2.3 °C during work shifts) and a very high humidity (mean RH 96%) in the cut-preparation area. Review of published evidence suggests that these conditions are likely to allow the virus to survive or remain stable for longer in the air, although our study could not test the stability of the virus in aerosols. These environmental conditions, if they are combined with a low fresh air ventilation rate and high levels of recirculation, may further increase the risk of inhalation transmission. The control measures observed in the production areas, such as visor wearing, wearing reusable fabric face coverings only when 2 m physical distance could not be maintained or using screens to separate workers who worked on the production lines for prolonged work shifts, might not be sufficient to reduce the potentially increased inhalation transmission risk [41].

### 4.5. Face Coverings

Face masks refer to surgical or respiratory masks. Face coverings include broader types and materials such as reusable cloth masks or simple scarves that cover the nose and mouth. In the outbreak site, face coverings (mostly reusable fabric masks) were mandatory in communal areas and on coaches, but visors were mandatory in the production areas. Wearing additional face coverings in the production areas became mandatory if physical distancing was not possible, and workers had to work face-to-face. There is substantial evidence that face masks and coverings will help to reduce emission of virus (source control) from infectious individuals and to protect others from exposure to droplets and large aerosols containing the virus [42]. A previous study demonstrated that SARS-CoV-2 had a better survival than the other viruses in the early stage (0~20 min) of leaving the infected individual [39]. As the virus could be highly infective in the early stages when leaving the infected individual, wearing face coverings or masks to minimise the risk at the source could be an effective measure. However, a laboratory study on source control using a manikin and cough simulator showed that a 3-ply cotton cloth face covering was only 51% effective at blocking emission of particles, whereas a face shield (also known as a visor) was only 2% effective [43,44]. Therefore, the mandatory visor wearing and wearing face covering for some activities in production areas of this outbreak site would be limited to partial source control.

The company provided staff with reusable fabric face coverings which were taken home to clean, as well as all types of face covering used, rather than washed in the general workwear laundry facility. A randomised control trial in hospital healthcare workers, focusing on seasonal respiratory viruses, found that those who self-washed their cloth masks by hand had double the risk of infection of those who used the hospital laundry and that double-layered cloth masks washed in the hospital laundry were as protective against respiratory infection as medical masks [45]. This suggests the protectiveness of the reusable fabric face covering would be better if they were washed in the factory laundry facility [46].

Visor-wearing alone would not protect workers from the potential risk of transmission in the production areas [43]. On the other hand, providing good quality face coverings or masks, both for source control and for protection against exposure, would be more effective than using visors alone in reducing the risk of the virus transmission. The effectiveness of these will rely on a careful risk assessment and workers being educated so that they can adhere to their proper use.

### 4.6. Surface Contamination

The proportion of positive surface samples (11.7% 7/60 samples) found in this outbreak site is similar to the positive rate of 8% found on high-touch surfaces in a community setting in the U.S., April to June 2020 [47]. However, the regular hygiene cleaning programme in the meat-processing plant might have reduced the likelihood of finding positive surface samples. In the outbreak site, the engineering workshop had the highest concentration of positive samples and the sample with the highest viral RNA load (Ct = 34.0) was also collected from this location. Two engineers tested positive for COVID-19, with one engineer testing positive four days before and the other engineer two days after the surface sampling, which may reflect the incidence in this location. However, these findings might only indicate the presence of infectious individuals in that environment and potential deficiencies in the surface cleaning rather than an indication that the workshop was a site of workplace transmission [19]. A 2020 sampling study assessed SARS-CoV-2 contamination of air and surface in a large Dutch meat-processing plant that was experiencing COVID-19 clusters. Although one-third of workers were tested positive, only 3% (six out of 203) of the surface samples collected were positive with low viral RNA loads (Ct > 38) [48].

### 4.7. Night-Shift Work

Similar to other studies of COVID-19 workplace outbreaks [6], the outbreak site had a higher infection attack rate among night-shift staff than among day-shift staff. This could be due to a combination of factors, such as the increased work demand, relaxed compliance to the COVID-19 control measures, potential health impact of the cold and damp work environment and health impact of circadian rhythm disruption when working nightshifts [3,49].

The first confirmed case in the day-shift staff was likely due to household transmission as the partner of this case, who also worked in the same factory but on nightshifts, tested positive four days prior. However, the outbreak in the day-shift staff did not develop much further.

Although there were some challenges identified, the observed COVID-19 control measures in the non-production areas were implemented relatively well. The company had an internal test and trace system led by senior staff, and the company also offered full pay for COVID-19-related self-isolation. The food safety controls already in place to reduce bacteria were also likely to be successful in reducing virus particles on surfaces and on hands.

### 4.8. Multi-Layered Control Measures

In summary, all routes of transmission may have played a part in this outbreak. Although there were challenges in their implementation, physical distancing, face coverings (including visors) and using physical barriers could have reduced the spray (droplet) transmission. Hand hygiene, extensive surface cleaning, wearing gloves and face coverings or visors (source control) could have reduced fomite transmission. However, the efficacy of the ventilation and the suitability of the face coverings/visor used could not be assessed adequately to support mitigating the potential inhalation transmission in the production areas. An epidemiological study involving a large number of meat-processing facilities in Nebraska, U.S., concluded that control measures such as mandatory face coverings and physical barriers would block mainly the larger respiratory droplets, and they would not fully protect workers against inhalation transmission, and the interventions should be accompanied by ventilation enhancements and worker education on face covering use and adherence [50].

There were challenges in the implementation of control measures in the meat-processing plant, including financial, production and reputational impacts on the employer, as well as workers’ worries about incomes or the incomes of the people they live with if they had to isolate. However, working with the local public health team, the company’s rapid testing, coupled with local contact tracing were likely to have been crucial in controlling the outbreak. Following the new measures, the cases in the workplace declined while the cases in the local community continued to increase rapidly. These are consistent with findings from previously studies [3,51,52].

Although our data could not reliably assess the adequacy of ventilation in the production areas of this workplace, it is well-demonstrated that effective ventilation can reduce risk of inhalation transmission [42]. If a refrigeration system in a multi-occupant space generates a high proportion of recirculated chilled air and cannot be adjusted to provide sufficient fresh air, suitable air cleaning interventions, such as HEPA (high-efficiency particulate air) filters and/or UV (ultraviolet radiation) air disinfection, could be used to remove or deactivate potential viruses from the recirculated air [5,42,53].

At this outbreak site, there was an extensive fogging programme. Currently no evidence supports the effectiveness of fogging in controlling SARS-CoV-2 transmission, but the financial costs and the environmental impact could be substantial. In comparison, the economic cost of face coverings could be relatively low and seems to be a cost-effective measure, as part of multi-layered controls, to contribute to reducing all routes of the virus transmission [42]. Nevertheless, evidence on the cost-effectiveness of various key SARS-CoV-2 control measures is mostly absent for supporting decision-making on control strategies and prioritisations when resources are limited.

Many different factors can contribute to a workplace outbreak. Therefore, a multi-layered approach is required to prevent and control outbreaks [54]. The choices and the priorities of the control measures should be informed by a workplace-specific risk assessment alongside scientific understanding of transmission routes. A visualising tool has been developed to help individuals and organisations understand the factors that influence transmission and the potential benefits of different mitigation measures, tackling the main transmission routes of SARS-CoV-2 [55].

### 4.9. Study Limitations

Our study has several limitations. The investigation was carried out as the outbreak was subsiding and was unable to conclusively establish transmission within the workplace. Therefore, the interpretation of the influence of different environmental factors, work activities and mitigation measures on the virus transmission is limited. The information on case numbers was based on non-exhaustive screening which limits the ability to establish complete chains of transmission between workers.

This study used CO_2_ measurements to assess the adequacy of ventilation provision in the occupied indoor workspaces. However, it was recognised that CO_2_ was also used in some of the meat packaging processes which affects the interpretation of the CO_2_ measurement results in the production areas. Therefore, no firm conclusion could be made on the adequacy of the ventilation in these areas. An alternative ventilation assessment approach should be considered in this situation, such as calculating an indicative air change rate from mechanical ventilation flow rates and room dimensions (where available or accessible). Given the right circumstances, the decay rate of CO_2_ (or a suitable tracer gas) can be used to estimate the ventilation rate [56]. Some knowledge of room occupancy and changes to factors such as windows and doors would be required.

Furthermore, information on the potential transmission risk factors and control measures were based on self-reports by company management, company records and study team members’ on-site observations. Therefore, this study cannot establish causal relationships between the risk factors investigated and the outbreak. Neither can this study assess conclusively the effectiveness of the control measures implemented. Nevertheless, it has identified potential transmission risk factors that were likely to have contributed to the outbreak.

## 5. Conclusions

Meat-processing plants appear to be at increased risk of outbreaks of COVID-19 occurring even with a wide range of control measures in place because of many work- and non-work-related risk factors, challenges in fully implementing the control measures and the challenges in the nature of the work. A high density of workers and an unfavourable mix of environmental conditions might play significant roles in the increased risk of virus transmission in the production areas. Further investigation of the production area ventilation effectiveness would be needed. Shared work transport is often required in these types of workplaces and could contribute to transmission within the workforce, even though this activity is difficult for employers to control.

This COVID-19 outbreak at the large meat-processing plant was likely contained by proactive leadership re-evaluating key controls and engaging with local health protection specialists. Leadership that encourages and monitors workers’ compliance with various control measures, including early identification of infectious cases through comprehensive workplace testing and contact tracing strategies, physical distancing, hand hygiene, surface cleaning, providing suitable sick pay policy and encouraging vaccination, are all important in the rapid response to control SARS-CoV-2 transmission or other similar respiratory infectious disease outbreaks in the workplace.

Our study illustrates the importance of site-specific risk assessments to inform choices made on the range of control measures, considering all potential routes of the virus transmission. The findings add to the existing literature on risk factors and control measures for COVID-19 outbreaks in meat-processing plants globally and contribute to future pandemic preparedness.

Meat-processing plants are part of the national essential infrastructure and could develop and rehearse, in collaboration with local health protection teams, an outbreak response plan for controlling future outbreaks of infectious diseases and to maintain business continuity.

## Figures and Tables

**Figure 1 ijerph-20-06806-f001:**
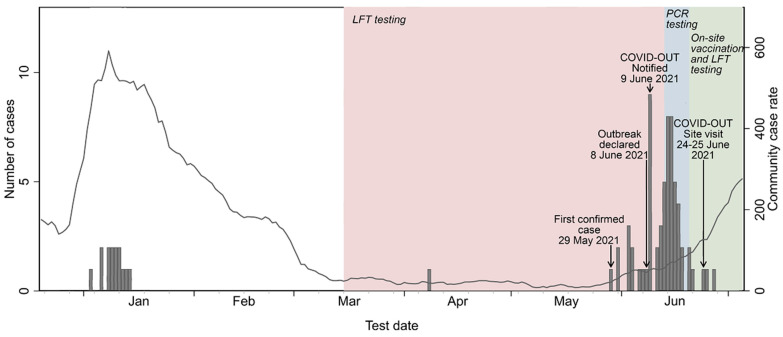
Timeline and epidemiological curve of the COVID-19 outbreak investigation in the meat-processing plant in 2021. Arrows indicate key dates of the outbreak and the COVID-OUT study. Bar charts represent the case numbers and dates provided by PHE (now UKHSA) and the company. Line charts represent the rolling 7-day case rate per 100,000 population for the lower tier local authority area (LTLA) of the site (contains public sector information licensed under the Open Government Licence V3.0) from Daily Summary of Coronavirus (COVID-19) in the UK, https://coronavirus.data.gov.uk/ ( accessed on 1 March 2022).

**Figure 2 ijerph-20-06806-f002:**
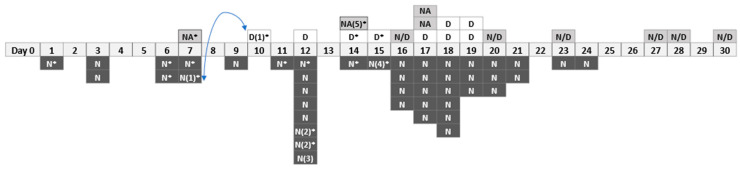
A timeline of confirmed COVID-19 cases, between 29 May 2021 and 27 June 2021, in the meat-processing plant. *****: S-gene positive which was a proxy for Delta variant of SARS-CoV-2 (information was available up to Day 15); D: Day-shift worker; N: Night-shift worker; NA: Unclear if the worker worked a dayshift or nightshift; N/D: Rotating dayshift and nightshift; Workers N(1) and D(1) were partners (shown by the arrow); Workers N(2) and N(2) were partners; Worker N(3): had a spouse diagnosed on the same day and a child diagnosed two days later; Worker N(4): had a relative diagnosed a day earlier; Worker NA(5): had a child diagnosed the same day and another child diagnosed two days later.

**Figure 3 ijerph-20-06806-f003:**
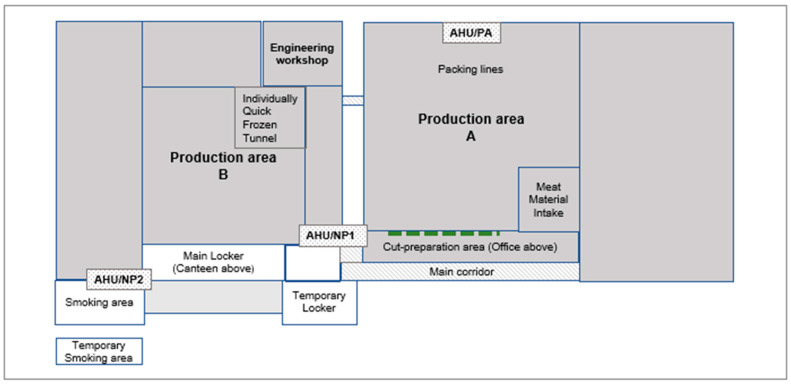
The building layout of the meat-processing plant (not to scale). The overall size of the site was about 43,000 m^2^ and the indoor area was 15,000 m^2^. Production area A was about 7000 m^2^ and production area B was about 4000 m^2^. AHU/PA: air handling unit in production area A; AHU/NP1: air handling unit for upstairs offices in main block; AHU/NP2: air handling units for canteen and locker rooms. In Production area A, a refrigeration system was operating and CO_2_ was used in the meat packaging process. The Cut-preparation area was connected to the main Production area A, separated only by flexible plastic drapes (shown by the dashed line).

**Figure 4 ijerph-20-06806-f004:**
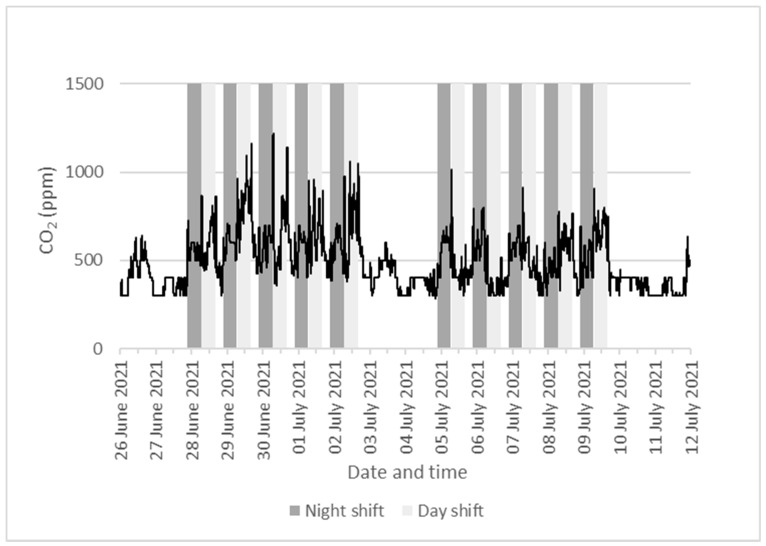
Real-time CO_2_ measurements in the main locker room of the meat-processing plant between 25 June 2021 and 11 July 2021. The line graph shows 30-min rolling average of CO_2_ concentrations. The dark grey area covers night-shift time (Sunday–Friday, 21:30–06:30) and the light grey area covers day-shift time (Monday–Friday, 07:00–16:00).

**Figure 5 ijerph-20-06806-f005:**
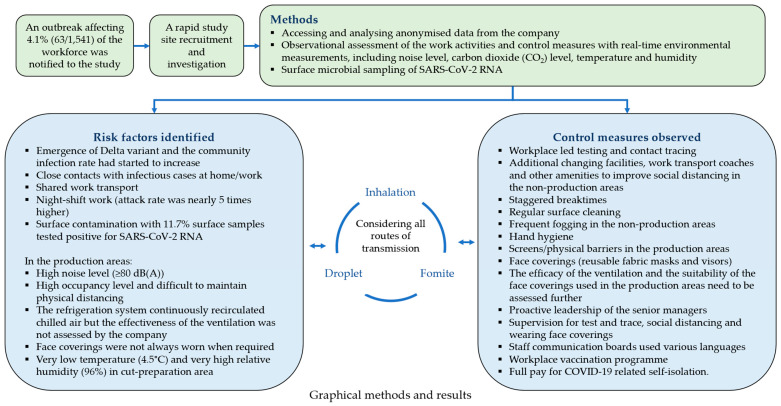
A graphical summary of the investigation of a COVID-19 outbreak in a large meat-processing plant in England, June 2021.

**Table 1 ijerph-20-06806-t001:** **SARS-CoV-2 RNA results of the 60 surface samples taken on 24 and 25 June 2021 from the meat processing plant following an outbreak**.

RT-PCR Results (From a Total of 60 Samples)			Level of RNA (Based on Ct Value)		
Confirm Positive	Suspected Positive	Negative	Moderate-High (Ct < 32.0)	Low (Ct 32.0–34.9)	Very Low-None (Ct ≥ 35.0 ^a^)
1 (1.7%)	6 (10.0%)	53 (88.3%)	0 (0.0%)	1 (1.7%)	59 (98.3%)
**Positive sample information**					
**Site area**	**Location in area**			**Mean Ct value ^b^**	**Estimated copies per cm^2 c^**
Engineering	Toolbox cupboard-door and 2 shelves			34.0	1007
Engineering	Toolbox cupboard-top of cupboard			37.9 ^d^	40
Engineering	Tool drawers			36.5 ^d^	240
Chiller 1	OCM screen			37.9 ^d^	14
Mission control	Dehumidifier			38.0 ^d^	34
Smoking shelter	Chair			37.5 ^d^	102
Canteen	Table top and seat			37.7 ^d^	38

^a^ Includes 53 samples with no SARS-CoV-2 RNA detected. ^b^ Mean Ct value for the N gene. ^c^ Extrapolation from copies per reaction to copies per sample collected based on the dilution factor, then divided by recorded sampling area. ^d^ Sample identified as suspected positive, defined as a sample with a single replicate testing positive for at least one target. The detection limit defined by the manufacturer was Ct 38.0. Abbreviations: Severe acute respiratory syndrome coronavirus 2 (SARS-CoV-2), Ribonucleic acid (RNA), Real-time polymerase chain reaction (RT-PCR), Crossing threshold (Ct) and Nucleocapsid (N).

## Data Availability

The data that support the findings of this study are available on reasonable request from the corresponding author. The data are not publicly available due to privacy or ethical restrictions.

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
