# Peer review of "A COVID-19 Outbreak in a Large Meat-Processing Plant in England: Transmission Risk Factors and Controls"

_ijerph, 2023, doi:10.3390/ijerph20196806_

Round 1

Reviewer 1 Report

Lack of originality: The abstract does not indicate any significant original contribution to the field of COVID-19 research or infectious disease control. The study appears to focus primarily on reporting the results of a specific outbreak in a meat processing plant.

     Limited scope: The study focuses on a single meat processing site in England. While it is important to understand outbreaks across different sectors, the generalizability of the results to the meat processing industry as a whole may be questionable.

     Limited methodology: The abstract mentions observations, real-time environmental measurements, and surface microbial sampling, but does not specify in detail the methods used. The lack of information about the methodology makes it difficult to assess the robustness of the study and replicate the results.

     Insufficient risk analysis: Although the summary mentions a specific risk assessment for the workplace, it is not clear how this assessment was carried out and what specific measures were implemented to prevent and control outbreaks of COVID-19. Without this information, it is difficult to determine the effectiveness of the aforementioned control measures.

     Limitations in Discussion of Results: The abstract does not provide an in-depth analysis of the results found. For example, while it is mentioned that 11.7% of surface samples tested were positive for SARS-CoV-2, the significance of these results in terms of disease transmission or effectiveness of control measures is not discussed.

Based on these reasons, I would consider declining the article due to lack of originality, limited scope, insufficient methodology, insufficient risk analysis, and limited discussion of results.

 minor revision.

Author Response

1. Lack of originality: The abstract does not indicate any significant original contribution to the field of COVID-19 research or infectious disease control. The study appears to focus primarily on reporting the results of a specific outbreak in a meat processing plant.

Response: We have provided additional information at the start of section 2. Materials and Methods (Page 2) and section 5. Conclusion (Page 15) to demonstrate the originality of our study. Our manuscript has described an investigation of a COVID-19 outbreak in a meat processing plant. This was one of the twelve on-the-ground investigations in a range of workplace settings, which we have conducted as part of the COVID-OUT study, to systematically evaluate workplace COVID-19 outbreaks. We developed and applied consistently the COVID-OUT study protocol that we have previously published (Reference 20 in our manuscript). Our study protocol was developed by combining a set of relevant World Health Organization (WHO) COVID-19 early investigation protocols and the lessons learnt from previous responses to global emerging pathogens, and adapting them to the investigations of workplace outbreaks in the UK. We have also conducted our own systematic review of international scientific literature on COVID-19 outbreaks in the workplace (Reference 13 in our manuscript). Our systematic review showed, when COVID-OUT was initiated, there was limited research based on direct observations of the workplace or real-time environmental measurements close to the time of workplace outbreaks. We could not find a study that had carried out workplace assessments of potential transmission risk factors and controls in a systematic manner as demonstrated in our study.

2. Limited scope: The study focuses on a single meat processing site in England. While it is important to understand outbreaks across different sectors, the generalizability of the results to the meat processing industry as a whole may be questionable.

Response: We were able to identify similar outbreak investigation studies in meat processing plants as part of our systematic literature review. Findings of this study at a single meat processing site in England was discussed in the context of the published studies internationally to provide greater insight on the virus transmission risk factors and control measures implemented in the meat processing industry. The real-world data collected from this single outbreak site will also support the interpretation of our other areas of work to identify and quantify COVID-19 outbreak risk across all sectors using linkages of national level data in England (relevant manuscripts are in preparation). The findings of our study will add to the existing literature on COVID-19 outbreak risk factors and control measures in the meat processing plants globally.

3. Limited methodology: The abstract mentions observations, real-time environmental measurements, and surface microbial sampling, but does not specify in detail the methods used. The lack of information about the methodology makes it difficult to assess the robustness of the study and replicate the results.

Response: In section 2. Materials and Methods, we have provided more detail information on how we developed our COVID-OUT study protocol (Reference 20 in our manuscript) which was applied consistently to the on-the-ground investigations across the twelve outbreak sites, including this particular meat processing plant. We have also provided more details on the methods used for collecting and testing the surface microbial samples. Subsections were also added to improve clarity on the specific methods used in the four main areas of the study, namely subsection 2.1. Outbreak identification and recruitment; 2.2 Case definition and identification; 2.3 Environmental assessment, including real-time environmental measurements; and 2.4. Surface microbial sampling.  

4. Insufficient risk analysis: Although the summary mentions a specific risk assessment for the workplace, it is not clear how this assessment was carried out and what specific measures were implemented to prevent and control outbreaks of COVID-19. Without this information, it is difficult to determine the effectiveness of the aforementioned control measures.

Response: Our study carried out an independent assessment of the control measures implemented two weeks after the outbreak was declared at this site. At that time, the outbreak was declining but there were still occasional positive cases identified in the workforce. The details of our assessments, including direct on-site observations of work activities and control measures, real-time environmental measurements and surface microbial sampling, were provided in section 2.Marterials and Methods. We were also able to access to the company’s own COVID-19 risk assessment, which allowed us to identify gaps in their assessment (e.g. no information was provided about the production area ventilation) as well as changes made in the control measures before and after the outbreak.  

5. Limitations in Discussion of Results: The abstract does not provide an in-depth analysis of the results found. For example, while it is mentioned that 11.7% of surface samples tested were positive for SARS-CoV-2, the significance of these results in terms of disease transmission or effectiveness of control measures is not discussed.

Response: We have provided an in-depth analysis of our results and organised our discussions in subsections, such as 4.2. Noise, 4.3. Ventilation, 4.4. Temperature and humidity, 4.5. Face coverings, 4.6. Surface contamination, 4.7. Night-shift work, 4.8. Multi-layered control measures, as well as 4.9. Study limitations. We have interpreted the proportion of positive surface samples (11.7%) appropriately and referenced findings of other similar studies. 

6. Based on these reasons, I would consider declining the article due to lack of originality, limited scope, insufficient methodology, insufficient risk analysis, and limited discussion of results.

Response: The information provided in above and the associated revisions made to the manuscript have now demonstrated that we have designed and carried out an original research that would add to the existing literature on risk factors and control measures for COVID-19 outbreaks in meat processing plants globally and would contribute to future pandemic preparedness. We have now provided sufficient methodology, risk analysis and discussion of results. 

Reviewer 2 Report

The undertaken area of the study is interesting, and yet, has not attracted sufficient attention from other scholars. The paper in generall is well written.

My detailed remarks are as follows:

1. In Introduction, please restate the objective 2 of the study, now it is too simplistic.

2. Section materials and methods requires significant adjustments. Please explain why this approach is the most suitable to acheive the objectives of this paper. Try to provide subsections. Explain in detail how was the study carried out.

3. In Conclusion section please think and provide implication of th study to various tatkholders.

Author Response

1. The undertaken area of the study is interesting, and yet, has not attracted sufficient attention from other scholars. The paper in general is well written.

Response: Thank you for your positive and encouraging comments.

2. In Introduction, please restate the objective 2 of the study, now it is too simplistic.

Response: We have added more specific information to the objective 2 at the end of the Introduction section (page 2).

3. Section materials and methods requires significant adjustments. Please explain why this approach is the most suitable to achieve the objectives of this paper. Try to provide subsections. Explain in detail how was the study carried out.

Response: We have explained in more detail on how we developed our COVID-OUT study protocol and why it would be suitable for achieving the study objectives at the start of section 2. Materials and Methods. We have also provided more details on the methods used for collecting and testing the surface microbial samples (Page 3-4). Subsections were also added to improve clarity on the specific methods used in the four main areas of the study, namely 2.1. Outbreak identification and recruitment; 2.2 Case definition and identification; 2.3 Environmental assessment, including real-time environmental measurements; and 2.4. Surface microbial sampling.  

4. In Conclusion section please think and provide implication of the study to various stockholders.

Response: We have provided more details in section 5. Conclusion on how our study would not only add to the global evidence base on risk and controls of COVID-19 in the meat processing plants but also inform businesses to improve their risk assessments and to help them to make right choices of control measures accordingly. Our study findings will also support future pandemic preparedness.

Reviewer 3 Report

Thank you for this overall nicely written and relevant manuscript. I think it is the right approach to discuss individual outbreaks against the background of the situation.

In the methods section, a short statistics section including the programs used would be helpful.

Overall, I would find it helpful if the CO2 measurements part was discussed at greater length. How could we make better use of the measurements? What alternative approaches would the authors suggest?

Throughout the discussion section, recommendations for action are often made from other studies. For example in line 528ff. I would recommend to rephrase or delete these sections and, as already written, to limit the discussion to how to achieve better results. 

The discussion of the mask part seems a bit long to me.

Author Response

1. Thank you for this overall nicely written and relevant manuscript. I think it is the right approach to discuss individual outbreaks against the background of the situation.

Response: Thank you for your positive and encouraging comments.

2. In the methods section, a short statistics section including the programs used would be helpful.

Response: We have added the relevant information in the Methods section, under subsections 2.2. Case definition and identification for producing the epidemic curve and 2.3. Environmental assessment for analysing the real-time measurement results.

3. Overall, I would find it helpful if the CO2 measurements part was discussed at greater length. How could we make better use of the measurements? What alternative approaches would the authors suggest?

Response: We have added the information in section 4.9. Study limitations, following the discussion on the potential alternative ventilation assessment approach.

4. Throughout the discussion section, recommendations for action are often made from other studies. For example in line 528ff. I would recommend to rephrase or delete these sections and, as already written, to limit the discussion to how to achieve better results.

Response: We have removed some of these discussions within subsection 4.8. Muti-layered control measures. However, we thought a lot of these discussions were useful to help businesses to make right choices of control measures, considering all potential routes of the virus transmission.   

5. The discussion of the mask part seems a bit long to me.

Response: We have shortened this section (4.5. Face coverings).

Round 2

Reviewer 1 Report

The graphics could be improved, using Rstudio, Biorender to make a graphical abstract and summarize the methodology and use a circus (graphic model) to illustrate the contaminations and dependencies of the environment.

Author Response

Further comments and suggestions

The graphics could be improved, using Rstudio, Biorender to make a graphical abstract and summarize the methodology and use a circus (graphic model) to illustrate the contaminations and dependencies of the environment.

Response: Thank you for your further suggestions to improve the manuscript. We have also communicated with the Academic Editor to clarify the journal’s expectation of this revision. We have now provided a graphical summary of the methods and results (see Figure 5) at the start of the Discussion section in page 10-11 of the manuscript, where the over study results were summarised.